# The Stress-Responsive microRNA-34a Alters Insulin Signaling and Actions in Adipocytes through Induction of the Tyrosine Phosphatase PTP1B

**DOI:** 10.3390/cells11162581

**Published:** 2022-08-19

**Authors:** Pierre-Jean Cornejo, Bastien Vergoni, Mickaël Ohanna, Brice Angot, Teresa Gonzalez, Jennifer Jager, Jean-François Tanti, Mireille Cormont

**Affiliations:** 1Université Côte d’Azur, Inserm, C3M, Team “Molecular and Cellular Physiopathology of Obesity and Diabetes”, 06204 Nice, France; 2Université Côte d’Azur, Inserm, C3M, “Team Microenvironnement, Signalisation et Cancer”, 06204 Nice, France; 3Aix Marseille Université, Inserm, INRAE, C2VN, 13385 Marseille, France

**Keywords:** obesity, insulin resistance, adipose tissue, GLUT4, VAMP2, NAD

## Abstract

Metabolic stresses alter the signaling and actions of insulin in adipocytes during obesity, but the molecular links remain incompletely understood. Members of the microRNA-34 (miR-34 family play a pivotal role in stress response, and previous studies showed an upregulation of miR-34a in adipose tissue during obesity. Here, we identified miR-34a as a new mediator of adipocyte insulin resistance. We confirmed the upregulation of miR-34a in adipose tissues of obese mice, which was observed in the adipocyte fraction exclusively. Overexpression of miR-34a in 3T3-L1 adipocytes or in fat pads of lean mice markedly reduced Akt activation by insulin and the insulin-induced glucose transport. This was accompanied by a decreased expression of VAMP2, a target of miR-34a, and an increased expression of the tyrosine phosphatase PTP1B. Importantly, PTP1B silencing prevented the inhibitory effect of miR-34a on insulin signaling. Mechanistically, miR-34a decreased the NAD^+^ level through inhibition of *Naprt* and *Nampt*, resulting in an inhibition of Sirtuin-1, which promoted an upregulation of PTP1B. Furthermore, the mRNA expression of *Nampt* and *Naprt* was decreased in adipose tissue of obese mice. Collectively, our results identify miR-34a as a new inhibitor of insulin signaling in adipocytes, providing a potential pathway to target to fight insulin resistance.

## 1. Introduction

White adipose tissue (WAT) by its metabolic and endocrine functions controls systemic metabolism. In WAT, adipocytes are responsive to insulin and adipocyte insulin sensitivity strongly influences whole body glucose and lipid homeostasis as well systemic insulin sensitivity [1]. Insulin signaling that regulates adipocyte metabolism is triggered by the activation of the tyrosine kinase activity of the insulin receptor (IR), which in turn phosphorylates on tyrosine residues the Insulin Receptor substrates (IRS) 1 and 2, leading to the activation of the AKT/PKB node, which plays a central role in the regulation of glucose uptake by promoting glucose transporter 4 (GLUT 4) translocation, lipogenesis, and lipolysis [1,2]. Obesity triggers WAT dysfunction, which is due to the induction of a variety of stresses within WAT, including low grade inflammation, hypoxia, oxidative and endoplasmic reticulum stress, and genotoxic stress, as recently described [3,4]. These stresses induce adipocyte insulin resistance [5,6] and several studies have shown that a disruption of insulin signaling in adipocytes causes whole body glucose intolerance and systemic insulin resistance [1]. However, the stress-signaling pathways involved in the alteration of insulin signaling and action in adipocytes are not yet fully identified.

Several processes are involved in the response to stress, including the induction of specific microRNAs (miR), which are small non-coding RNAs of around 20–22 nucleotides [7]. Each miR regulates genetic programs through the modulation of several target mRNAs. Some miRNAs are involved in the control of the metabolic homeostasis and numerous studies pinpointed a deregulation in miR expression in several metabolic organs, including WAT, in obesity and insulin resistance states [8,9,10]. Adipocytes are an important source of circulating miRs that can modulate insulin sensitivity of other organs [11], and some miRNAs were shown to modulate WAT inflammation and adipocyte differentiation [12,13,14,15]. However, miR regulating insulin signaling in adipocytes in response to adipose stresses are poorly defined even if miR-103/107 were reported to be involved in the downregulation of the insulin receptor by targeting caveolin-1 [16].

Members of the miR-34 family (miR-34a, b, and c) are induced by a stress response program [17,18], and are the most prevalent p53-induced miRNAs [19]. They are tumor suppressors; in addition, they are involved in organ development and aging, and are deregulated in several diseases [19,20]. Several studies have shown that the expression and activity of p53 are increased in WAT and adipocytes of obese mice [4,21,22], together with an increase in DNA damage in adipocytes [4]. Moreover, the activation of adipocyte p53 has been highlighted as an important signaling pathway leading to adipocyte and systemic insulin resistance, WAT inflammation, and metabolic reprogramming of adipocytes during obesity [4,22,23].

In parallel to p53, miR-34a-5p (hereafter referred as miR-34a) is also upregulated in WAT during obesity and was found to inhibit WAT browning and to promote WAT inflammation by inhibiting M2 macrophage polarization [24,25]. Interestingly, it was shown that the NAD+-dependent protein deacetylase Sirtuin-1 (SIRT1) and the v-SNARE vesicle-associated membrane protein 2 (VAMP2) are direct targets of miR-34a in different cell types [26,27]. These two proteins are involved in the regulation of insulin signaling and GLUT4 translocation [28,29].

However, whether elevated miR-34a in adipocytes has a functional role in the inhibition of insulin signaling and action remains unknown Therefore, in this study, we wanted to determine whether miR-34a was involved in the alteration of insulin signaling and insulin-induced glucose transport in adipocytes, by studying adipocytes overexpressing miR-34a, and by increasing its expression in mice fat pads.

## 2. Materials and Methods

### 2.1. Material

Anti-VAMP2 antibodies were from Synaptic Systems (Göttingen, Germany). Monoclonal antibodies against phosphotyrosine were from Becton Dickinson France SAS (Grenoble, France). Polyclonal antibodies against PTP1B were from R&D systems (Minneapolis, MN, USA). Polyclonal antibodies directed against acetylated lysine (pan-acetyl lysine) were from Millipore. Polyclonal antibodies against GLUT4, GLUT1, insulin receptor β subunit and monoclonal antibodies against HSP90 were from Santa Cruz Biotechnology (Santa Cruz, CA, USA). Phosphospecific antibodies against Akt (Threonine-308 or Serine-473), extracellular signal regulated kinase (ERK)1/2 (Thr-202/Tyr204), and antibodies against Akt, ERK1/2 and Sirt1 were from Cell Signaling Technology (Beverly, MA, USA). Antibodies against α tubulin were from Sigma-Aldrich (St. Louis, MO, USA). Horseradish peroxidase-coupled anti-species antibodies were from Jackson ImmunoResearch Laboratories Europe (Suffolk, UK). Control miRNA and miRIDIAN miR-34a mimic (C-310529-07-0002), and the SMARTpool ON-TARGETplus siRNA against *Ptp1b* were purchased from Dharmacon (Horizon Discovery).

### 2.2. Animals

*Ob/ob* male mice (B6.Cg-Lep^ob^/J, JAX^®^ Mice Strain) and the heterozygote mice *ob/+* from the colony as well as C57BL/6J male mice were purchased from Charles River Laboratories (St. Aubin les Elbeuf, France) for the studies. Mice were maintained on a 12 h light/12 h dark cycle with free access to water and diet. *Ob*/*ob* and *ob*/+ mice were studied at the age of 11 weeks. Seven-week-old male C57BL/6J mice were fed with a chow diet (NCD, 8% of calories from fat) or a high-fat diet (HFD, 60% of calories from fat, SSNIFF, Soest, Germany) for 13 weeks, as described [4]. Intraperitoneal Glucose Tolerance Test (IPGTT) was performed after starvation for 6 h. Glucose (1 g D-glucose/kg body weight) was administrated by intraperitoneal injection in awake mice. Blood was collected via tail vein at different time points, and glucose levels were measured using a Glucometer (Medisens Optimum XCD, Abbott, Rungis, France), as described [4]. 

### 2.3. In Vivo Injection of miR-34a Mimic into Fat Pads

Male C57BL/6J mice (12 weeks-old, *n* = 10) were anesthetized, and one epididymal fat pad was injected with a control miRNA (50 µL, 170 µM in NaCl 0.9%) while the other one was injected with a miR-34a mimic (50 µL, 170 µM in NaCl 0.9%) with a syringe with a needle of 22 G. Epididymal adipose tissues were removed 24 h later, in the morning, freeze-clamped in liquid nitrogen and stored at −80 °C until used. The Principles of Laboratory Animal Care (NIH publication “Guide for the care and use of laboratory animals” publication no.85-23, revised 1985 were followed, as well as the European Union guidelines on animal laboratory care (https://ec.europa.eu/environment/chemicals/lab_animals/index_en.htm, accessed on 2 August 2022). All procedures were approved by the Animal Care Committee of the Université Côte d′Azur.

### 2.4. Preparation of Adipocytes and the Stromal Vascular Fraction from Adipose Tissue

Epididymal adipose tissues (eAT) from *ob/+* mice and *ob/ob* mice or chow and high-fat diet fed mice were subjected to collagenase digestion to obtain adipocytes and the stromal vascular fraction (SVF) as described [30]. Briefly, finely cut eAT pieces were incubated for 45 min at 37 °C with collagenase (1 mg/mL) in Krebs-Ringer medium, buffered with bicarbonate and HEPES (KRBH: 120 mM NaCl, 4.8 mM KCl, 2.5 mM CaCl_2_, 1.2 mM KH_2_PO_4_, 1.2 mM MgSO_4_, 15 mM NaHCO_3_, 30 mM HEPES, pH7.4), and containing 0.3 mM glucose and 1% BSA. After digestion, eAT cell suspension was filtered through nylon screen (pore size 250 mesh) in 50 mL polypropylene conical tube. The medium under the floating adipocytes was collected and centrifuged (1000× *g* 10 min) to obtain a cell pellet corresponding to the SVF. The floating adipocytes were washed three times by flotation with KRBH without BSA. The cell pellet corresponding to the SVF was washed by resuspension in KRBH without BSA followed by centrifugation. The adipocytes and SVF were stored at −80 °C for further analysis.

### 2.5. 3T3-L1 Adipocytes Culture and Transfection

The 3T3-L1 fibroblast was obtained from ATCC (Cat#CL-173TM) and was cultured in Dulbecco′s Modified Eagle Medium (DMEM) supplemented with 10% (*v*/*v*) newborn calf serum (Life Technologies, St. Louis, MO, USA), 2 mmol/L glutamine, 100 units/mL penicillin and 100 units/mL streptomycin until confluence. Two days after confluence, cells were induced to differentiate into adipocytes in DMEM supplemented with 10% (*v*/*v*) fetal bovine serum (FBS) containing isobutyl methylxanthine 0.5 µM (Sigma-Aldrich Inc., St. Louis, MO, USA, Cat# I5879), dexamethasone 0.25 µM (Sigma-Aldrich Inc., St. Louis, MO, USA, Cat# D1756), rosiglitazone 10 µM (Enzo Life Science, Farmingdale, NY, USA, Cat# ALX-350-125) and insulin 800 nmol/L (Humulin, Lilly, Indianapolis, IN, USA, Cat# HI0290). After two days, cells were cultured with DMEM 10% FBS containing 800 nmol/L insulin for two additional days [31]. Cells were used between day 2 and 7 after the end of the differentiation protocol when the adipocyte phenotype appeared in more than 90% of the cells. The 3T3-L1 adipocytes were transfected with miR-34a mimic or control mimic (5–10 nM), or indicated siRNA (20 nM) with scramble siRNA as control by using INTERFERin^TM^ (Ozyme, St. Quentin en Yvelines, France) as described [32], and cells were harvested 48 h later for analysis.

### 2.6. Glucose Transport

Glucose transport in 3T3-L1 adipocytes was measured as described previously [33]. Briefly, 3T3-L1 adipocytes were serum starved for 18 h and incubated without or with insulin at the concentration indicated in the legend of figure for 20 min in Krebs-Ringer phosphate buffer (10 mm phosphate buffer (pH 7.4), 1.25 mm MgSO_4_, 1.25 mm CaCl_2_, 136 mm NaCl, 4.7 mm KCl). Glucose transport was determined by the addition of 2-[3H] deoxyglucose (0.1 mM, 0.5 µCi/mL) for 3 min at 37 °C. Then, the cells were washed four times with ice-cold PBS to stop the reaction. Cells were lysed and glucose uptake was assessed by scintillation counting. Results were normalized for protein content by using bicinchoninic acid (BCA) protein assay reagent (ThermoFisher Scientific, Waltham, MA, USA).

### 2.7. Western Blot

The 3T3-L1 adipocytes were solubilized in ice-cold buffer containing 20 mmol/L Tris pH 7.5, 150 mmol/L NaCl, 1% Triton X-100, 2 mmol/L orthovanadate, 100 mmol/L NaF, 10 mmol/L Na_4_P_2_O_7_ and Complete protease inhibitor cocktail (Roche Diagnostics, Meylan, France). Snap-frozen epididymal adipose tissues were homogenized by using Precellys tissue homogenizer in the same buffer. After 1 h at 4 °C, lysates were centrifuged at 14,000× *g* rpm for 10 min at 4 °C, and the protein concentration was determined using BCA protein assay reagent. Immunoblotting was performed, as previously described [33,34], using between 10–25 µg of proteins that were separated on 7.5–10% SDS-PAGE gels and transferred to polyvinylidene difluoride membranes. Membranes were blocked with Tris-buffered saline-Tween (TBST) containing 4% non-fat dried milk and incubated overnight at 4 °C with the indicated primary antibodies at the dilution recommended by the manufacturer. After incubation with horseradish-peroxidase conjugated secondary antibodies diluted at 1:5000, proteins were detected by enhanced chemiluminescence (Pierce ECL Western Blotting Substrate, ThermoFisher Scientific, Waltham, MA, USA), and a PXi4 GeneSys imaging system. Densitometric data analysis was performed using MultiGauge software.

### 2.8. RNA Isolation, RT-PCR and Quantitative PCR

RNAs were prepared from 3T3-L1 adipocytes or snap-frozen epididymal adipose tissues or adipocytes using TRIzol^TM^ reagent (Life Technologies, Villebon-sur-Yvette, France) and RNeasy total RNA kit, according to the manufacturer’s instructions (Qiagen, Courteboeuf, France). Isolated RNA was quantified with NanoDrop spectrophotometer and was reverse transcribed using Transcriptor first strand cDNA synthesis kit (Roche Diagnostics, Meylan, France). Realtime quantitative PCR was performed with sequence detection systems (StepOne Plus Real-time PCR system, ThermoFisher Scientific Inc, Waltham MA, USA) and Fast SYBR Green Master Mix. Primers for real-time PCR Primers were designed using Primer Express software (Applied Biosystems, Austin, TX, USA) and synthetized by Eurogentec (Seraing, Belgium). The list of primers used will be provided under request. Each primer couple was validated by characterizing the slopes of Ct over a range of three log dilutions and the curve of temperature dissociation of the amplicons. Gene expression values were calculated based on the comparative cycle threshold Ct method (2−ΔΔCt). Levels of mRNA were normalized to the expression value of the housekeeping gene Rplp0 (36B4) and were presented relative to the control group. The expression of mature miR-34a was quantified by using the dedicated kit from Applied Biosystems (ThermoFisher Scientific Inc., Waltham, MA, USA). The level of miR-34a was expressed relative to the invariant small RNA sno208 and presented relative to the control group.

### 2.9. NAD^+^ Measurement

NAD^+^ measurement levels were measured by enzymatic reactions using the NAD^+^/NADH quantification kit according to the manufacturer’s instructions (Sigma-Aldrich Inc., St. Louis, MO, USA, Cat# MAK037-1KT).

### 2.10. Statistical Analysis

All results are expressed as means ± SEM. Statistical analysis was performed using GraphPad Prism software (GraphPad Software Inc., La Jolla, CA, USA). Statistical significance of differences between two groups of mice was evaluated with the Mann–Whitney test and a comparison between adipocytes expressing miR-34a or not was performed with Student’s *t* test. Comparison between multiple groups was evaluated using one-way ANOVA with post hoc Bonferroni’s test analysis. Time-dependent evolution of glycemia in the IPGTT was analyzed by two-way ANOVA. Correlation analysis was carried out by Spearman’s test. *p* value ≤ 0.05 was considered significant.

## 3. Results

3.1. miR-34a Expression Is Increased in Adipose Tissues of Diet-Induced Obese Mice and Genetically Obese Mice

We first determined whether the expression of miR-34a was changed in different adipose tissue depots of diet-induced and genetically obese mice compared to their lean counterpart. As excepted, *ob*/*ob* mice and HFD-fed mice had a higher body weight (Appendix A) and had an increase in glycemia (Appendix A) compared to the *ob/+* and chow-fed mice, respectively. We evaluated glucose metabolism in chow-fed and HFD-fed mice by conducting IPGTT, as excepted HFD-fed mice had reduced glucose tolerance compared to chow-fed mice (Appendix A). 

The expression level of miR34a was increased in epididymal and subcutaneous adipose tissues (eAT and scAT, respectively) of HFD-fed mice by 1.8- and 4-fold, respectively (Figure 1A). By contrast, HFD led to a reduction in the miR-34a level in brown adipose tissue (BAT). Expression of miR-34a was also 2.2 higher in eAT of genetically obese *ob/ob* mice compared to their lean *ob/+* counterpart (Figure 1B). The increased expression of miR-34a in eAT of HFD-fed mice and the higher expression of miR-34a in eAT of chow-fed mice compared to scAT (Figure 1A) are in agreement with a previous study [25]. In eAT, HFD led to an increase in the miR-34a level in the adipocyte fraction but not in the stroma vascular fraction (SVF), whereas an induction of miR-34a was observed both in the adipocyte and SVF fractions of eAT from genetically obese *ob/ob* mice (Figure 1B,C).

Because miR-34a is a transcriptional target of p53 whose expression is increased in adipose tissue of obese mice [4,21,22], we searched for a correlation between miR-34a and the *Cdkn1a* mRNA level, a transcriptional target of p53. The mRNA level of *Cdkn1a* and the miR-34a level were positively correlated in eAT and adipocytes from eAT as well as in scAT (Figure 1D–F). By contrast, no correlation was found in BAT (data not shown).

### 3.2. The Overexpression of miR-34a in Adipocytes Inhibits Insulin-Induced Glucose Transport and VAMP2 Expression

Because miR-34a expression is increased in the adipocytes of obese mice, we determined the consequences of miR-34a overexpression on insulin signaling and actions in adipocytes. To this aim, 3T3-L1 adipocytes were transiently transfected with a mimic of miR-34a or a control miRNA and then incubated without or with insulin (0.5, 1, 10 nM) before measurement of deoxyglucose (DOG) uptake. The overexpression of miR-34a reduced insulin-induced glucose uptake both at a maximal and submaximal concentration of insulin (Figure 2A). This inhibition occurred without any changes in the expression of GLUT4 or GLUT1 (Figure 2B,C). Strikingly, the expression of the v-SNARE protein VAMP2, which is involved in the fusion of GLUT4-containing vesicles with the plasma membrane [35], and described as miR-34a target in pancreatic β cells [26], was decreased in 3T3-L1 adipocyte overexpressing miR-34a (Figure 2D) as well as in the adipose tissue of HFD-fed mice (Figure 2E), while its mRNA expression level was not significantly reduced (Figure 2F).

### 3.3. The Overexpression of miR-34a in Adipocytes Inhibits Insulin-Induced Akt Activation

Because miR-34a inhibited insulin-induced glucose uptake at a submaximal concentration of insulin, we examined whether miR-34a affected the activation of AKT by insulin [1,2]. Western blotting analysis demonstrated that 3T3-L1 adipocytes overexpressing miR-34a mimic displayed a 40% decrease in phosphorylation of AKT on Threonine 308 (Thr 308) in response to insulin stimulation (0.5 nM) compared to cells treated with a control miRNA (Figure 3A). The insulin-induced phosphorylation of AKT on Serine 473 (Ser473) was also decreased in adipocytes overexpressing miR-34a (Appendix A). To determine whether the increased expression of miR-34 observed in adipose tissue of obese mice could have a functional role in the downregulation of AKT activation, we overexpressed miR-34a in eAT of lean mice by injecting a miR-34a mimic in one eAT depot, whereas the other depot was injected with a miRNA negative control. Random-fed mice were sacrificed 24 h after injection to analyze miR-34a expression and AKT phosphorylation. Strikingly, the phosphorylation of AKT in the random-fed state was decreased by 40% in the eAT depot injected with the miR-34a mimic compared to the depot injected with the miRNA negative control (Figure 3B), and as expected, the injection of the mimic led to an increase in miR-34a expression (Figure 3C).

### 3.4. miR-34a Inhibits Insulin Signaling by Promoting the Expression of the Tyrosine Phosphatase PTP1B

To further dissect how miR-34a inhibits AKT activation, we examined the proximal insulin signaling in 3T3-L1 adipocytes. Notably, miR-34a overexpression significantly decreased the insulin-induced tyrosine phosphorylation of the β subunit of the IR and of IRS proteins (Figure 4A). In line with the lower level of tyrosine phosphorylation of IR and IRS, the protein expression and the mRNA level of the tyrosine phosphatase PTP1B, which dephosphorylate both IR and IRS proteins [36], were significantly increased following overexpression of miR-34a in 3T3-L1 adipocytes (Figure 4B–D).

To address whether the inhibition of insulin signaling in adipocytes overexpressing miR-34a was due to elevated PTP1B, we treated adipocytes overexpressing miR-34a with a siRNA targeting *Ptp1b* (si-*Ptp1b*). The siRNA against *Ptp1b* slightly but significantly reduced basal PTP1B protein expression and totally blunted the induction of PTP1B expression induced by miR-34a (Figure 5A,B). The downregulation of VAMP2 was not prevented by the siRNA against *Ptp1b* (Figure 5A). By contrast, the blockade of PTP1B upregulation markedly prevented the decrease in the insulin-induced tyrosine phosphorylation of IR, IRS, and AKT phosphorylation evoked by the miR-34a overexpression (Figure 5B).

### 3.5. miR-34a Promotes the Expression of PTP1B through Sirtuin-1 Inhibition

We next interrogated how miR-34a regulated the expression of PTP1B. The expression of the *Ptp1b* gene can be repressed through deacetylation of histones by the histone/protein deacetylases Sirtuin-1 (SIRT1), a described target of miR-34a in several cell types [27,37]. Thus, an increase in PTP1B expression in 3T3-L1 adipocytes overexpressing miR-34a could be due to a change in histone acetylation. Consistently, overexpression of miR-34a led to elevated acetylation of histones (Figure 6A), and notably, this increase was comparable to that obtained following the knockdown of SIRT1 (Figure 6B). The overexpression of miR-34a did not significantly change the protein expression of SIRT1 (Figure 6B,C) but it reduced the amount of NAD^+^ (Figure 6D). Since SIRT1 is a NAD^+^-dependent deacetylase, these findings support the notion that miR-34a acts by decreasing SIRT1 catalytic activity rather than its expression in 3T3-L1 adipocytes. To further confirm that inhibition of SIRT1 induced by miR-34a could be involved in the increased expression of PTP1B and the alteration in insulin signaling, we evaluated the effect of SIRT1 silencing. Notably, downregulation of SIRT1 led to an upregulation of PTP1B expression (Figure 6E), which was comparable to that obtained by the miR-34a overexpression (Figure 4B). Meanwhile, the knockdown of SIRT1 reduced the insulin effect on the tyrosine phosphorylation of IR and IRS as well as on the phosphorylation of AKT on threonine 308 (Figure 6F). Taken together, these data support that the increase in miR34a, by inhibiting SIRT1, promotes the expression of PTP1B, thereby leading to an inhibition of insulin signaling.

### 3.6. miR-34a Overexpression Decreases the Expression of Nampt and Naprt mRNA

The nicotinamide phosphoribosyltransferase (NAMPT) and nicotinate phosphoribosyltransferase (NAPRT) are key enzymes in the NAD^+^ biosynthetic pathways, which allow the synthesize of NAD^+^ from nicotinamide or nicotinic acid [38,39]. We, thus, investigated whether the decrease in NAD^+^ induced by elevated miR-34a expression was due to an alteration in the expression of these two enzymes. *Nampt* mRNA was already identified as a direct target of miR-34a in different cell types [40,41,42]. Accordingly, overexpression of miR-34a in 3T3-L1 adipocytes led to a significant reduction in the level of *Nampt* mRNA (Figure 7A). We analyzed the sequence of the *Naprt* mRNA and found a sequence in the 3′UTR matching with the miR-34a seed sequence (Figure 7B), and the overexpression of miR-34a induced a marked decrease in the level of *Naprt* mRNA (Figure 7C). Consistent with these results, the mRNA level of *Naprt* was markedly decreased in eAT of HFD-fed mice (Figure 7D), while a decrease in the expression of *Nampt* mRNA was already reported in AT of both mice and patients with obesity [43,44].

## 4. Discussion

In this study, we provide compelling evidence that miR-34a, whose expression is increased in adipose tissue and adipocyte during obesity, contributes to the alteration in insulin signaling and action in adipocytes. Our results demonstrate that the overexpression of miR-34a in adipocytes downregulates the expression of VAMP2 which is necessary to increase the GLUT4 level at the plasma membrane in response to insulin [35] and inhibits the insulin signaling by increasing the expression of the tyrosine phosphatase PTP1B. Mechanistically, miR-34a reduces NAD^+^ levels possibly through the targeting of NAMPT and NAPRT, which may lead to the inhibition of SIRT1 deacetylase activity. Our results suggest that the decrease in SIRT1 activity induced by miR-34a promotes PTP1B expression by enhancing histone acetylation (Figure 8).

Consistent with a previous report [25], we observed an upregulation of miR-34a in both epididymal and subcutaneous WAT of HFD-fed mice, and we extended this observation to genetically obese mice. In contrast to miR-34a, we did not observe an increased expression of the other members of the family miR-34b and c (data not shown). The increase in miR34a expression was observed in the adipocytes of the HFD-fed mice and the genetically obese *ob/ob* mice. In the SVF, miR-34a expression is increased in genetically obese *ob/ob* mice, but not in diet-induced obese mice. The reasons for such differences are not known but can be due to a higher level of inflammatory or genotoxic stresses in the SVF of the *ob/ob* mice due to greater obesity in *ob/ob* mice compared to HFD-fed mice. In humans, previous studies have reported a positive association between the expression of adipose miR-34a and BMI [45,46], the degree of insulin resistance, and the metabolic inflammation [25]. Moreover, the expression of miR-34a has also been reported to be negatively associated with *peroxisome proliferator activated receptor γ PPARγ* expression in visceral WAT of patients with obesity [47]. We found that the regulation of miR-34a is completely different between WAT and BAT during obesity, with a downregulation of miR-34a in BAT of obese mice. The reasons for such a difference in the regulation of miR-34a between these two types of adipose tissues are presently unknown. It could be an adaptive mechanism to preserve BAT functions because miR-34a was shown to inhibit the brown fat gene program by downregulating fibroblast growth factor (FGF) receptor 1and beta Klotho, two components of the FGF21 receptor complex [24].

The elevated expression of the stress-responsive miR-34a in WAT during obesity is consistent with the well-known induction of a variety of stresses within obese WAT [2,3,4]. Indeed, the expression of miR-34a is upregulated in adipocytes by inflammatory cytokines such as tumor necrosis factor α (TNFα) or by palmitic acid, suggesting that inflammatory and lipotoxic stresses within WAT can contribute to the increase in miR-34a expression [25]. miR-34a is also transcriptionally regulated by p53 [19]. We, and other researchers, have reported that p53 activity is increased in adipocytes during obesity [4,21,22] which can be due to an increase in DNA damage in obese adipocytes [4]. It is, thus, plausible that DNA damage through activation of p53 can also induce miR-34a. Accordingly, we found that the elevated expression of miR-34a in obese WAT is positively correlated with the expression of p53 targets. Therefore, miR-34a can be an integrator of metabolic and genotoxic stress within adipocytes during obesity.

Previous studies have shown that the biology of adipose tissues can be affected by miR-34a in several ways. Downregulation of miR-34a using lentivirus expressing antisense miR-34a in obese mice markedly reduced all WAT depots by promoting WAT browning and BAT functions. This was in part due to the blockade of the miR-34a inhibitory effect on the FGF21-receptor complex, FGFR1 and beta klotho [24]. A recent study using adipocyte-specific miR-34a knockout mice showed that miR-34a is secreted by adipocytes in exosomal vesicles and transported in adipose macrophages where it repressed their anti-inflammatory M2 polarization by targeting Krüppel-like factor 4. Therefore, through paracrine action on macrophages, adipocyte-secreted miR-34a promotes WAT inflammation, insulin resistance and metabolic dysregulation [25]. In addition to this paracrine mechanism, we highlighted here that miR-34a can act directly in adipocytes to inhibit insulin signaling and insulin-induced glucose uptake. It is likely that the inhibition of the effect of insulin on glucose uptake induced by the miR-34a overexpression is due to the decrease in insulin signaling and to the inhibition of GLUT4 incorporation into the plasma membrane. Indeed, even if we have not assessed the insulin-induced GLUT4 incorporation into the plasma membrane, the inhibition of VAMP2 expression can inhibit this process. Indeed, VAMP2 is the v-SNARE associated with the intracellular vesicles containing GLUT4 and it was shown to be indispensable for the fusion of these vesicles with the plasma membrane and to insulin-induced glucose transport [35,47]. Moreover, we did not observe any change in the expression of GLUT4 and GLUT1 in adipocytes following miR-34a overexpression, which could have been another mechanism by which miR-34a would have decreased glucose uptake. This downregulation of VAMP2 is consistent with studies showing that VAMP2 is a direct target of miR-34a in different cell types, including pancreatic beta cells [26,48,49]. Interestingly, it is plausible that miRs negatively control the expression of VAMP2 in adipocytes during obesity because its expression was markedly reduced in eAT of obese mice without significant change in its mRNA level. Whether miR-34a is one of such miRs remains to be established.

The activation of the IR tyrosine kinase by insulin and the subsequent AKT stimulation also play a critical role in the regulation of glucose transport and lipid metabolism in adipocytes [1]. Our results established that miR-34a overexpression markedly reduced AKT activation by insulin in 3T3-L1 adipocytes. Moreover, we showed that the overexpression of miR-34a in fat pads of lean mice is sufficient to decrease the level of AKT phosphorylation in the random-fed state, suggesting an important role for elevated miR-34a in obese adipocytes for the attenuation of AKT activation. This inhibition of AKT was linked with a decrease in the activation of IR tyrosine kinase and with reduced tyrosine phosphorylation of IRS proteins. Several potential inducers of miR-34a such as inflammatory cytokines, saturated fatty acids, or DNA damage and activation of p53 inhibit insulin signaling in adipocytes [4,5,25]. Therefore, miR-34a can be a key player mediating the inhibition of insulin signaling by several stressors of adipocytes. Because miR-34a overexpression markedly blunted the tyrosine phosphorylation of both IR and IRS proteins, we hypothesized that miR-34a could regulate the expression of tyrosine phosphatases. Protein-tyrosine phosphatases such as the transmembrane (receptor-like) leukocyte antigen related phosphatase (LAR) and the non-transmembrane tyrosine phosphatase PTP1B are key negative regulators of insulin signaling by dephosphorylating IR and or IRS proteins [50]. The expression of PTP1B is increased in adipocytes during obesity [51,52], and our findings suggest that an elevated miR-34a expression might contribute to this increase. Importantly, *Ptp1b* siRNA experiments support that the inhibitory effect of miR-34a on proximal insulin signaling is dependent on the increase in PTP1B expression. Several studies have shown that PTP1B is a major negative regulator of insulin signaling in muscles or the liver [53,54], but its importance in adipocytes is still a matter of debate. Indeed, invalidation of PTP1B in adipocytes does not improve insulin signaling and whole-body glucose homeostasis [55]. In contrast and in line with our findings, a reduction in the PTP1B expression in liver and adipose tissue of obese mice by antisense oligonucleotide treatment increases the IR tyrosine phosphorylation and AKT activation [56]. Furthermore, overexpression of PTP1B in 3T3-L1 adipocytes decrease the insulin-induced tyrosine phosphorylation of IR and IRS1, and PI3K activation, but the insulin effect on glucose uptake was not modified [57]. Therefore, the inhibitory effect of miR34a overexpression on insulin-induced glucose uptake might result from its action on both the insulin signaling cascade through PTP1B induction and from the targeting of VAMP2 limiting GLUT4 incorporation in the plasma membrane.

It was previously shown that SIRT1 can repress *Ptp1b* gene transcription at the chromatin level by promoting the deacetylation of histones [37]. *Sirt1* is a direct target of miR-34a in different cell types, and we found a huge increase in histone acetylation together with an increase in *Ptp1b* mRNA following the overexpression of miR-34a in adipocytes. However, the expression of SIRT1 was not significantly inhibited by the overexpression of miR-34a, but the level of NAD^+^ was decreased. Likewise, miR-34a reduces NAD^+^ level and SIRT1 activity in hepatocytes [41]. The balance between NAD-consuming reactions and NAD-biosynthetic pathways controls the NAD level. NAMPT and NAPRT are involved in the first steps of NAD biosynthesis from nicotinamide and nicotinic acid, respectively. The expression of NAMPT and the level of NAD^+^ are reduced in WAT during obesity, and NAMPT deficiency in adipocytes markedly alters the endocrine function of WAT and the insulin sensitivity [43,44,58]. NAMPT was shown to be a direct target of miR-34a in hepatocytes [41], and accordingly, the overexpression of miR-34a markedly reduced the *Nampt* mRNA level in adipocytes. Moreover, our data indicate that miR-34a also negatively regulates *Naprt* mRNA and this inhibition could be mediated by a direct binding of miR-34a because the analysis of the *Naprt* mRNA sequence revealed a potential binding site. Therefore, it is likely that elevated miR-34a in adipocytes decreases NAD^+^ level in adipocytes by directly targeting both NAMPT and NAPRT, resulting in a decrease in SIRT1 activity which promotes PTP1B expression through an increase in histone acetylation. Nonetheless, we cannot exclude the possibility that activation of other inhibitory mechanisms such as c-Jun N-terminal kinase (JNK) activation and IRS1 serine phosphorylation, due to the decrease in SIRT1 activity by miR-34a, also mediate the downregulation of insulin signaling [59,60].

In conclusion, our study supports that the stress responsive miR-34a is an inhibitor of insulin signaling and of insulin-stimulated glucose uptake in adipocytes by acting on the NAMPT-NAPRT/SIRT1/PTP1B axis and by repressing the expression of the GLUT4 vesicles associated v-SNARE VAMP2. The combination of paracrine [25] and cell autonomous action of miR-34a is likely to synergize to induce adipocyte insulin resistance and metabolic dysregulation. Therefore, miR-34a appears to be a potential target to improve WAT dysfunction, which is an important determinant of obesity-induced metabolic dysregulation.

## Figures and Tables

**Figure 1 cells-11-02581-f001:**
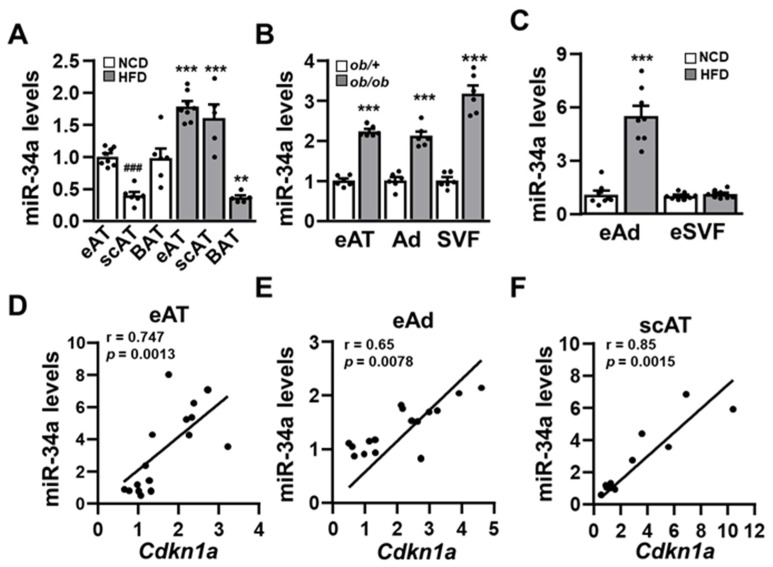
miR-34a expression is increased in adipose tissues and adipocytes of obese mice: Relative expression level (qPCR) of miR-34a (**A**) in epididymal white adipose tissue (eAT), subcutaneous white adipose tissue (scAT), and brown adipose tissue (BAT) of mice (*n* = 5–8) fed with normal chow diet (NCD) or a high-fat diet (HFD) for 13 weeks; (**B**) in eAT, adipocytes (Ad) and stromal vascular fraction (SVF) of genetically obese *ob/ob* mice and their lean *ob/+* littermates (*n* = 6), and (**C**) in adipocytes and eSVF of eAT isolated from mice on NCD or HFD for 13 weeks. (**D**–**F**) Correlation analysis of miR-34a and *Cdkn1a* mRNA levels in the indicated white adipose tissue or adipocyte fraction. ANOVA with Bonferroni’s multiple comparisons test was used for A and B. Correlation was assessed by non-parametric Spearman’s test. Bar graphs are mean values ± SEM. ** *p* < 0.01 and *** *p* < 0.001 NCD vs. HFD; ### *p* < 0.001 scAT vs. eAT. Relative levels of miR-34a and *Cdkn1a* mRNA were normalized to sno208 and *Rplp0* (*36B4*), respectively.

**Figure 2 cells-11-02581-f002:**
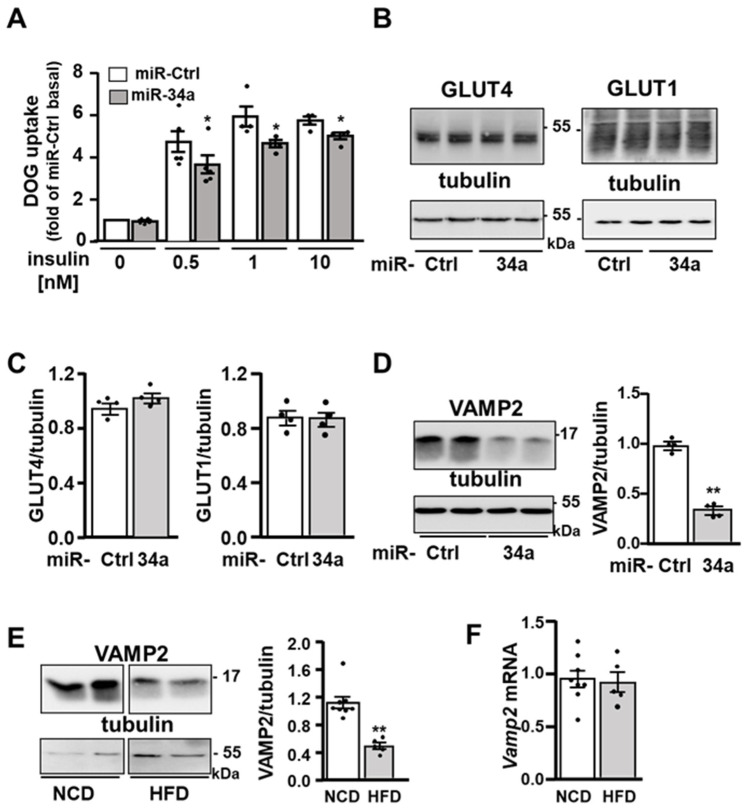
miR-34a overexpression inhibits insulin-induced glucose transport and reduces VAMP2 protein expression: (**A**) 2-deoxyglucose (DOG) uptake in 3T3-L1 adipocytes transfected with a control miRNA (miR-Ctrl) or miR-34a mimic and treated without or with the indicated concentration of insulin. Bar graph represents mean values ± SEM of four independent experiments. (**B**) Representative immunoblot (**left** panel) of the expression of GLUT4 and GLUT1 with tubulin as loading control in 3T3-L1 adipocytes transfected with a control miRNA (miR-Ctrl) or miR-34a mimic. (**C**) Bar graph represents the densitometry analysis for the GLUT4 or GLUT1/tubulin ratio from four independent experiments. (**D**) Representative immunoblot (**left** panel) of the expression of VAMP2 with tubulin as loading control in 3T3-L1 adipocytes transfected as in (**B**). Bar graph (**right** panel) represents the densitometry analysis for the VAMP2/tubulin ratio from four independent experiments. (**E**) Representative immunoblot (**left** panel) of the expression of VAMP2 with tubulin as loading control in eAT of mice fed with a normal chow diet (NCD) or a high-fat diet (HFD) for 13 weeks. Bar graph (**right** panel) represents the densitometry analysis for the VAMP2/tubulin ratio in eAT of *n* = 5–8 mice/group. (**F**) Relative level of *Vamp2* mRNA in eAT of mice fed with a NCD or HFD for 13 weeks (*n* = 5–8 mice/group). Relative level of *Vamp2* mRNA was normalized to *Rplp0* (*36B4*). Bar graphs are mean values ± SEM. * *p* < 0.05 and ** *p* < 0.01.

**Figure 3 cells-11-02581-f003:**
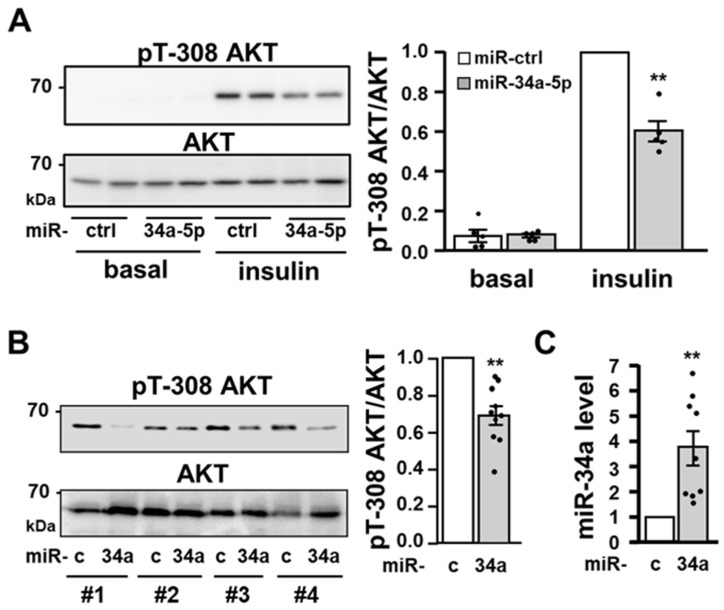
The overexpression of miR-34a in 3T3-L1 adipocytes or in epididymal fat pads of mice inhibits AKT activation: (**A**) Representative immunoblot (**left** panel) of the level of phosphoThr308-AKT (pT-308 AKT) and total AKT in 3T3-L1 adipocytes transfected with a control miRNA (miR-Ctrl) or a miR-34a mimic and treated without or with insulin (0.5 nM) for 10 min. Bar graph (**right** panel) represents the densitometry analysis for the pT-308 AKT/AKT ratio from five independent experiments. (**B**) Representative immunoblot (**left** panel) of the level of pT-308 AKT and total AKT in epidydimal fat pads of the same mice in the random-fed state injected with a control miRNA (miR-c) or a miR-34a mimic (34a), as described in the Material and Methods section (fat pads from 4 mice out of 10 were shown). Bar graph (**right** panel) represents the densitometry analysis for the *p*-T308 AKT/AKT ratio in the epididymal fat pads of the *n* = 10 mice. (**C**) Relative expression level (qPCR) of miR-34a in the epididymal fat pads of the mice (*n* = 9). The relative level of miR-34a was normalized to sno208. Bar graphs are mean values ± SEM. ** *p* < 0.01.

**Figure 4 cells-11-02581-f004:**
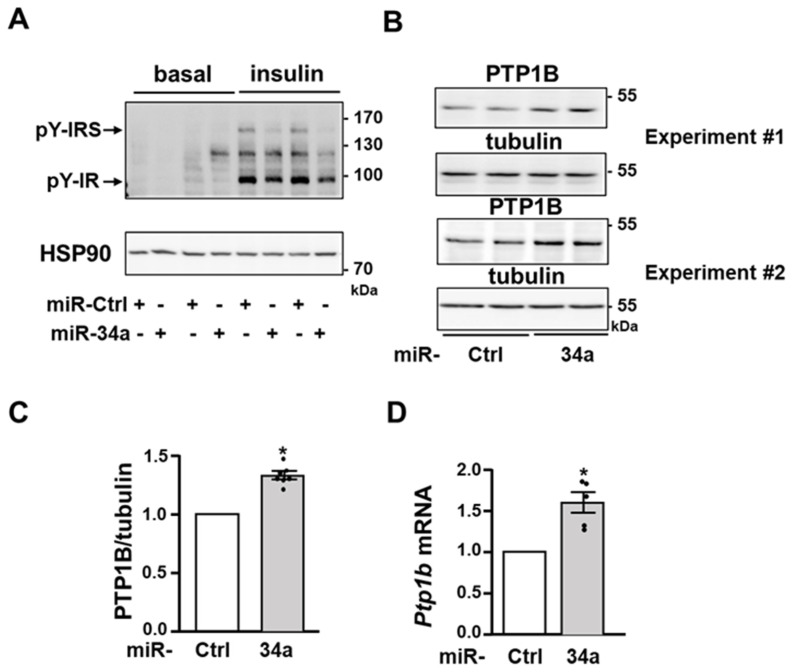
The expression of the tyrosine phosphatase PTP1B is increased in adipocytes overexpressing miR-34a: (**A**) Representative immunoblot of the level of tyrosine phosphorylation (pY) of IR and IRS and expression of AKT (loading control) in 3T3-L1 adipocytes transfected with a control miRNA (miR-Ctrl) or a miR-34a mimic, and treated without (basal) or with insulin (0.5 nM) for 10 min. (**B**) Immunoblot of PTP1B expression with tubulin as loading control in unstimulated 3T3-L1 adipocytes transfected as in (**A**). Immunoblots from two independent experiments are shown. (**C**) Densitometry analysis for the PTP1B/tubulin ratio (*n* = five independent experiments). (**D**) Relative expression level (qPCR) of *Ptp1b* mRNA normalized to *Rplp0* (*36B4*). Bar graphs are mean ± SEM. * *p* < 0.05.

**Figure 5 cells-11-02581-f005:**
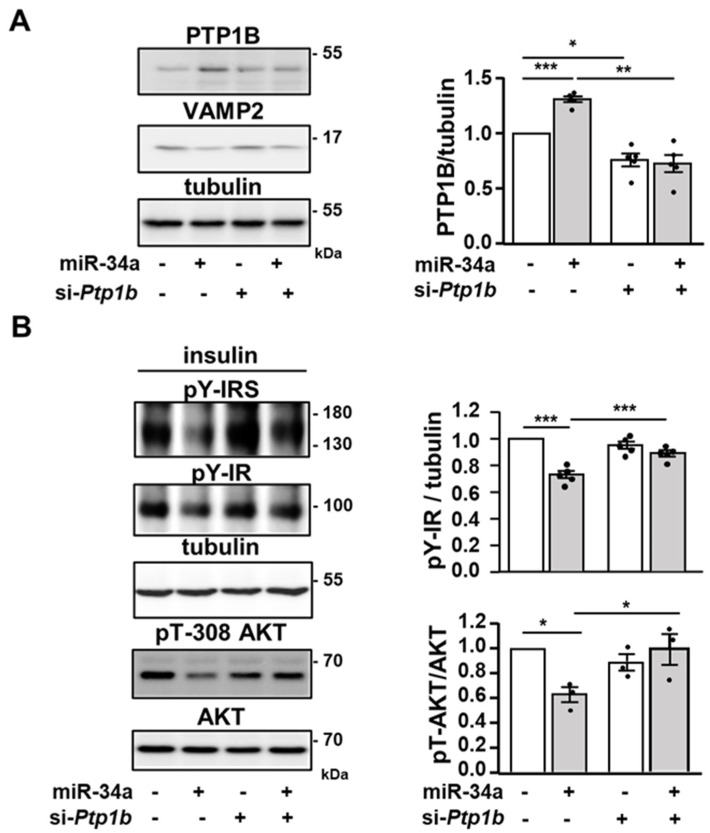
miR-34a inhibits insulin signaling by increasing PTP1B expression: (**A**) Representative immunoblot (**left** panel) of the expression of PTP1B, VAMP2, and tubulin as loading control in 3T3-L1 adipocytes transfected with a control miRNA or a miR-34a mimic in combination with a siRNA against *Ptp1b* or a control siRNA. Bar graph (**right** panel) represents mean values ± SEM of the densitometry analysis for the PTP1B/tubulin ratio from five independent experiments. (**B**) Representative immunoblot (**left** panel) of the level of tyrosine phosphorylation (pY) of IR and IRS, phosphorylation of AKT on threonine 308 (pT-308 AKT), and expression of total AKT and tubulin (loading control) in 3T3-L1 adipocytes transfected as in (**A**) and treated with insulin (0.5 nM) for 10 min. Bar graphs (**right** panel) represent the densitometry analysis for the pY-IR/tubulin and pT-308 AKT/AKT ratio. Data are mean values ± SEM of 3–5 independent experiments. * *p* < 0.05, ** *p* < 0.01 and *** *p* < 0.001.

**Figure 6 cells-11-02581-f006:**
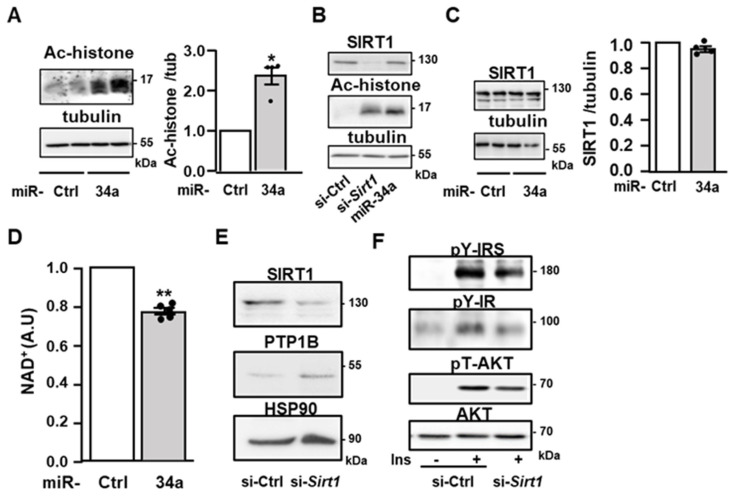
miR-34a increases histone acetylation and decreases NAD^+^ level: (**A**) Representative immunoblot (**left** panel) of the level of acetylated histones (Ac-histone) detected with a pan-acetyl lysine antibody in 3T3-L1 adipocytes transfected with a control miRNA (miR-Ctrl) or a miR-34a mimic. Tubulin was used as loading control. Bar graph (**right** panel) represents the densitometry analysis for the acetylated histones/tubulin ratio. Data are mean values ± SEM of four independent experiments. * *p* < 0.05. (**B**) Representative immunoblot of the expression of SIRT1, acetylated histones detected with an antibody against pan-acetyl lysine, and HSP90 as loading control in 3T3-L1 adipocytes transfected with a siRNA against *Sirtuin-1* (*Sirt1*) or a control siRNA (si-Ctrl), or with a miR-34a mimic. A representative immunoblot of two independent experiments is shown. (**C**) Representative immunoblot (**left** panel) of the expression of SIRT1 with tubulin as loading control in 3T3-L1 adipocytes transfected as in (**A**). Bar graph (**right** panel) represents the densitometry analysis for the SIRT1/tubulin ratio. Data are mean values ± SEM of four independent experiments. (**D**) NAD^+^ level in 3T3-L1 adipocytes expressed relative to the cells transfected with control miRNA. Data are mean values ± SEM of four independent experiments. ** *p* < 0.01. (**E**) Representative immunoblot of the expression of SIRT1, PTP1B and HSP90 (loading control) in 3T3-L1 adipocytes transfected with a control siRNA (si-Ctrl) or a siRNA against *Sirt1*. (**F**) Representative immunoblot of the level of the tyrosine phosphorylation (pY) of IR and IRS, phosphorylation of AKT on threonine 308 (pT-AKT), and expression of AKT in 3T3-L1 adipocytes transfected as in (**E**) and treated without or with insulin (Ins, 0.5 nM) for 10 min.

**Figure 7 cells-11-02581-f007:**
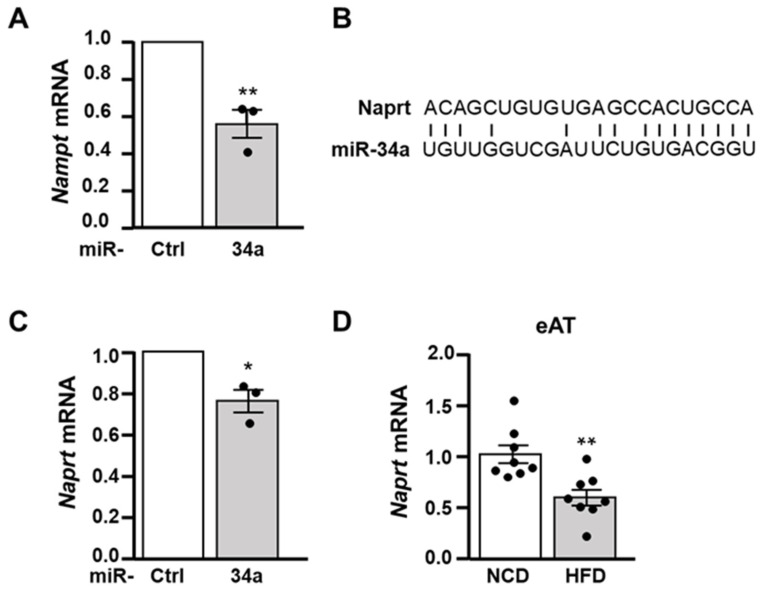
miR-34a decreases the expression of *Nampt* and *Naprt* mRNA. (**A**) Relative mRNA level of *Nampt* in 3T3-L1 adipocytes transfected with a miR-34a mimic or a control miRNA (*n* = 3). (**B**) Alignments of the miR-34a sequence with the putative miR-34a binding site in the 3′UTR of mouse *Naprt* mRNA. (**C**) Relative mRNA level of *Naprt* in 3T3-L1 adipocytes transfected with a miR-34a mimic or a control miRNA (*n* = 3). (**D**) Relative mRNA level of *Naprt* in epidydimal AT (eAT) of mice fed with a NCD or a HFD for 13 weeks (*n* = 8/group). Bar graphs are mean values ± SEM. * *p* < 0.05 and ** *p* < 0.01.

**Figure 8 cells-11-02581-f008:**
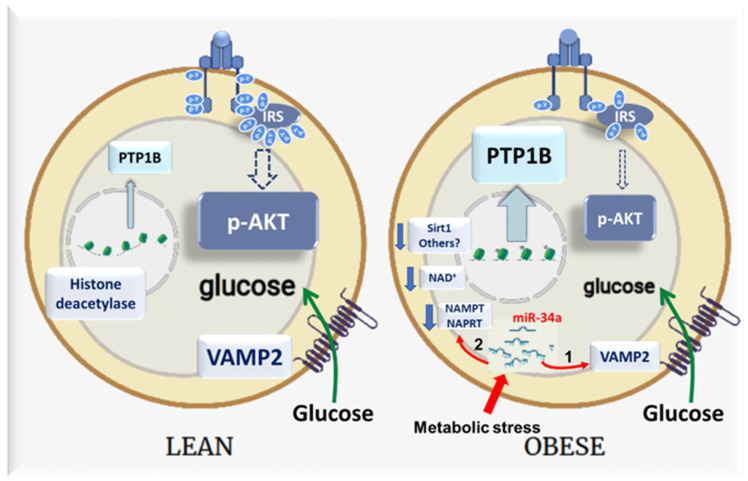
Schematic models illustrating the inhibitory mode of action of miR-34a on insulin signaling and glucose uptake in adipocytes during obesity. The miR-34a expression in adipocyte is low in lean state, insulin efficiently induces insulin receptor (IR) and insulin receptor substrates (IRS) tyrosine phosphorylation, leading to AKT phosphorylation and activation. Consequently, insulin increases glucose transport in adipocytes. In obese conditions, metabolic stresses increase the expression of miR-34a in adipocytes. The increase in miR-34a expression (**1**) inhibits the expression of the v-SNARE VAMP2 (vesicle associated membrane protein 2) involved in the fusion of GLUT4-containing vesicles with the plasma membrane and (**2**) inhibits the expression of NAMPT and NAPRT, leading to a decrease in NAD^+^ level, which results in an increase in acetylation of histones, possibly due to the inhibition of Sirt1 activity, a NAD^+^-dependent deacetylase. This is correlated with an increase in the expression of the tyrosine phosphatase PTP1B, which plays a pivotal role in the decrease in the tyrosine phosphorylation of IR and IRS induced by miR-34a, leading to a decrease in AKT activation. The defect in insulin signaling together with the decrease in VAMP2 expression are responsible for defective glucose transport in response to insulin in adipocytes overexpressing miR-34a.

## Data Availability

Not applicable.

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
