# Peer review of "The Stress-Responsive microRNA-34a Alters Insulin Signaling and Actions in Adipocytes through Induction of the Tyrosine Phosphatase PTP1B"

_cells, 2022, doi:10.3390/cells11162581_

Round 1

Reviewer 1 Report

In this interesting paper Cornejo et al. aimed to investigate the role of miR-34a in adipocytes' insulin signaling. They have demonstrated that the overexpression of miR-34A induces the expression of PTP1B by enhanced histone acetylation triggered by decreased Sirt1 activity. My primary concern is that the authors do not discuss or highlight the possible role of miR-34a on IRS serine phosphorylation. 

1Why are miR-34a levels increased in ob/ob SVF and not in eSVF from HFD?

Overall, methods must be better described. For example, how adipocytes from mice adipose tissue were obtained, counted, and differentiated from other lipid droplets?

 Please provide further information regarding IRS phosphorylation. IRS-1 or IRS-2? 

4Line 108: "Obese mice were glucose 108 intolerant as determined by IPGTT." Were is this result throughout the paper? 

5When first mentioned, please define the abbreviations. 

6Authors propose that the overexpression of miR34a in adipocytes inhibits insulin-induced GLUT4 (Fig.2). However, membrane GLUT4 fraction should be provided once only total expression was provided.

Line 93." antibodies against Akt (Threonine-308 or Serine-473). Please state why only Th308 phosphorylated AKT is presented in the results section, while no data about Ser473 phosphorylated AKT is presented.

 Individual data points should be presented in the figures instead of solid bars.

 The authors should provide all the original blots used to obtain the data presented in the figures. The authors conclude that the elevated expression of PTP1B supports the inhibitory effect of miR-34a on insulin signaling. However, there is no data regarding the activity of PTP1B. Further information should be provided (i.e., immunoprecipitation assays).

Reviewer 2 Report

1. Materials and Methods: Animal’s part needs to be rewritten giving the following suggestions/recommendations:

-Breeding/generation of ob/ob and their ob/+ littermates need to be described with details for genotyping, primers, etc.

-Lines 101-109 need to be moved to the Results with the putative title: Weight and metabolic characteristics of mice. Also, the results for IPGTT need to be included there.

 2. Results: Fig.1-on my opinion, ANOVA (either one way or two ways) is the more appropriate statistical analysis, at least for parts A and B.

3. Results: Fig.2A-Does the overexpression of miR-34a have an effect on the basal levels of glucose uptake? Authors can put the corresponding bars on the graph.

    Fig.2B-Authorts showed the unaltered total level of Glut4. Instead, plasma membrane (PM) of  Glut 4 need to be shown as indication of insulin-induced Glut4 movement to PM.

    Fig.3B-Authors presented basal phosphorylation levels of pTAkt-308 in the epidydimal fat pads after mir34a injection. To make conclusion regarding the insulin signaling they should also show the insulin-dependent pTAkt-308.

   Fig.4B-it will be clearer to readers if authors indicate that the parts are 2 independent experiments directly on the Figure.

4. Discussion: Lines 462-464-authors should state that “mir34a in fat pads of lean mice is sufficient to decrease basal AKT phosphorylation”.

The scheme showing the proposed mechanisms of p53-dependent miR-34a regulation of insulin signaling will be very helpful for the readers.

Reviewer 3 Report

This is a comprehensive original research paper investigating a role of stress-responsive microRNA-34a in insulin signaling pathway in adipocytes thus providing a new potential therapeutic target for insulin resistance.

In the last paragraph of Introduction section the aim of this study should be clearly stated, not obtained results. Obtained results should be stated in the first paragraph of the Discussion section.

The Results section should contain only results, not explanations of why certain molecule has specific effects, This should be moved to discussion section. For example: "To understand how miR-34a overexpression perturbs the stimulation of glucose 239 transport in response to insulin, we searched for miR-34a validated targets published in 240 the literature which could explain such effect. Interestingly the SNARE protein VAMP2, 241 involved in the fusion of GLUT4-containing vesicles with the plasma membrane [35], was 242 described as miR-34a target in pancreatic  cells [36]. "

"Because miR-34a overexpression markedly blunted the tyrosine 297 phosphorylation of both IR and IRS proteins, we hypothesized that miR-34a could regu- 298 late the expression of tyrosine phosphatases. PTP1B is a tyrosine phosphatase which acts 299 as an inhibitor of the insulin signaling pathway decreasing the phosphorylation of the 300 insulin receptor and the IRS [37]."

"This finding supports that the inhibitory effect of miR-34a on insulin signaling 322 is dependent on the elevated expression of PTP1B."

Overall this paper is well written with celarly described methodology as well as results, and presented evidence support drawn conclusions.

Round 2

Reviewer 2 Report

I recommend to:

1 Put WT/ND or WT/HFD for Suppl.Fig2B, D and E for clarity;

2. Put labels Experiment 1" and "Experiment 2" on the Suppl. Figure 2, as well as indicate that the top part is the Ser473-P Akt for Experiment 1.

Author Response

Dear reviewer,

We modified the supplemental figures 1 and 2 as you requested